# The Identification of Goat *KCNJ15* Gene Copy Number Variation and Its Association with Growth Traits

**DOI:** 10.3390/genes15020250

**Published:** 2024-02-17

**Authors:** Jiahao Zhao, Zhe Liu, Xianwei Wang, Xiaoling Xin, Lei Du, Huangqing Zhao, Qingming An, Xiaoting Ding, Zijing Zhang, Eryao Wang, Zejun Xu, Yongzhen Huang

**Affiliations:** 1College of Animal Science and Technology, Northwest A&F University, Xianyang 712100, China; mr.jiahaomd@foxmail.com (J.Z.); liuzhesci@126.com (Z.L.); duleisci2021@163.com (L.D.); zhaohuangqing@nwafu.edu.cn (H.Z.); dxtsci@126.com (X.D.); 2Henan Provincial Animal Husbandry General Station, Zhengzhou 450008, China; wangxianwei2008@163.com; 3Institute of Animal Husbandry and Veterinary Science, Henan Academy of Agricultural Sciences, Zhengzhou 450002, China; xinxl726@163.com (X.X.); vincezhang163@163.com (Z.Z.); eryaowang@outlook.com (E.W.); 4College of Agriculture and Forestry Engineering, Tongren University, Tongren 554300, China; anqingming2009@163.com

**Keywords:** association analysis, growth traits of goat, *KCNJ15*, copy number variation (CNV), quantitative polymerase chain reaction (qPCR)

## Abstract

(1) Background: Copy number variation (CNV) is a critical component of genome structural variation and has garnered significant attention. High-throughput screening of the *KCNJ15* gene has revealed a correlation between the CNV region and the growth traits of goats. We aimed to identify the CNV of the *KCNJ15* gene in five goat breeds and analyze its association with growth characteristics. (2) Methods: We utilized 706 goats from five breeds: Guizhou black goat (GZB), Guizhou white goat (GZW), Bohuai goat (BH), Huai goat (HH), and Taihang goat (TH). To evaluate the number of copies of the *KCNJ15* gene using qPCR, we analyzed the correlation between the CNV and growth characteristics and then used a universal linear model. The findings revealed variations in the distribution of different copy number types among the different goat breeds. (3) Results: Association analysis revealed a positive influence of the CNV in the *KCNJ15* gene on goat growth. In GZB, individuals with duplication types exhibited superior performance in terms of cannon bone circumference (*p* < 0.05). In HH, individuals with duplication types exhibited superior performance in terms of body slanting length (*p* < 0.05). Conversely, normal TH demonstrated better body height and body weight (*p* < 0.05), while in GZW, when CN = 3, it performed better than other types in terms of body weight and chest circumference (*p* < 0.05). However, in BH, it had no significant effect on growth traits. (4) Conclusions: We confirmed that the CNV in the *KCNJ15* gene significantly influences the growth characteristics of four distinct goat breeds. The correlation between *KCNJ15* gene CNVs and goat growth traits offers valuable insights to breeders, enabling them to employ precise and efficient breeding methods that enhance livestock welfare, productivity, and overall economic benefits in the industry.

## 1. Introduction

China has rich genetic resources of livestock and poultry, and many excellent breeds have been selected via China’s diverse natural environment and artificial selection, creating roughage tolerance, a high reproduction rate, and a strong ability to adapt to the environment. Goats are among the earliest animals domesticated by humans and serve as a significant source of meat, milk, and fur. However, with the development of science and technology, agriculture has gradually become mechanized and modernized, and the use of goats has gradually shifted to serving meat. Changes in native goats are already evident. As the goat genome has been progressively studied, genetic inheritance and variation have been used to improve goat reproduction and production. After long-term selection and cultivation, the meat yield, quality, and other meat production properties of modern goat breeds are higher than those of original breeds. Crossbreeding often leads to increased productivity in goats. China boasts the largest goat population globally, and there is a growing trend in the consumption of goat meat as part of the daily diet. Thus, it is imperative to employ biotechnological breeding techniques to enhance the size of goats, consequently boosting their meat production capacity. Leveraging molecular breeding techniques for livestock identification presents a promising avenue for enhancing overall livestock production practices.

Guizhou black and white goats are mainly produced in the Bijie area of Guizhou province. They are locally protected varieties and have the characteristics of excellent meat quality, large size, and strong stress resistance. Due to the special ecological environment and harsh natural selection in which they live, they have a strong ability to survive and have a unique set of body structure and physiological mechanisms to withstand roughness, moisture, and cold. Taihang goats are produced in the bordering areas of the Shanxi, Hebei, and Henan provinces on the east and west sides of Taihang Shandong. With a strong constitution and medium build, this breed has good production performance and can be used for meat, cashmere, and leather. The terrain in the Taihang Mountains is high and complex, which makes Taihang goats good at climbing. Huai goats are an ancient breed with a history of more than 1000 years. They are medium in size, have short and dense hair, are precocious, reproduce quickly, like dryness and dislike moisture, love wrestling, and are easy to raise. Their coat is basically all white, and both the males and females have horns. Due to their fast growth and short growth cycle, they are mainly used for the production of skins and wool, especially board skin and goat hair. They have enjoyed a long and enduring reputation in domestic and foreign markets, and their economic value is higher than that of many other goat breeds. Bohuai goats are a cross between the Huai goat and the Boer goat. The whole body of these goats is covered in white fur, and the fur from the neck to the head is brown. Some dorsal columns have a colored backline. After more than 20 years of crossbreeding and improvement, Bohuai goats have not only retained the characteristics of Huai goats, with good meat quality and a high reproduction rate, but also have the advantages of Boer goats, with fast growth, high meat yield, high slaughter rate (48–60%), and high meat production. These goats have good performance, tender meat, low fat content, and a high lean meat rate, and they have good economic and social benefits.

Copy number variations (CNVs) are closely associated with biological signs and properties and play an important role in biological evolution. High-throughput screening of the *KCNJ15* gene has revealed a correlation between the CNV region and the growth traits of goats [1]. There are many mechanisms for the formation of CNVs [2]. Understanding genetic differences among individuals is facilitated by the study of CNVs, which also plays an integral role in the phenotypic polymorphism of mammals. Additionally, CNVs can serve as a crucial candidate molecular marker for economic traits [3]. After a large number of screenings, CNVs, which are related to the important traits of growth and development, have greatly increased the proportion of excellent livestock individuals, providing strong proof for animal breeding [4]. Quantitative trait loci (QTLs) include DNA regions related to specific phenotypic traits, which can be attributed to polygene effects in different degrees [5]. The characteristics of growth and development show continuous variation due to the interference of external environmental factors [6]. QTL mapping technology can directly link gene variation with growth traits and can provide reference for selection. Via our query on the Online Mendelian Inheritance in Man (OMIM) platform, we discovered that *KCNJ15* is a member of the subfamily J of the potassium inward rectifying channel. This channel is designated as *KCNJ15*, and its corresponding entry on OMIM can be found at https://omim.org/entry/602106, accessed on 20 September 2023. Currently, the majority of research in this field focuses on investigating *KCNJ15*’s role in various medical diseases and its involvement in the human immune system.

The *KCNJ15* protein produced by the *KCNJ15* gene acts as a pivotal part of the inward regulation of the potassium channel, the voltage valve ion channel, the potassium channel, and potassium ion binding [7,8,9]. Studies have shown that the *KCNJ15* gene does not play a direct causal role in the development of the hornless phenotype in goats. However, it is possible that *KCNJ15* interacts with the *ERG* and *FOXL2* genes, thereby influencing the presence or absence of horns in goats [10]. Simon et al. found that the *KCNJ15* gene is present in all 334 genotypes of hornless goats of different breeds, and thus, it can be used as an accurate genetic diagnosis for hornless goats [11]. Concurrently, previous research has demonstrated the involvement of *KCNJ15* in the regulation of insulin secretion [12], as well as Alzheimer’s disease by modulating immune-related pathways [13].

Currently, there is limited literature on the relationship between the *KCNJ15* CNV and goat growth traits. Our comprehension of its function and mechanism is still in its early stages, and there is a lack of studies investigating the correlation between the variation in the *KCNJ15* gene copy number and the growth characteristics of Chinese goats. In our study, we utilized sequencing technology to specifically identify the region of copy number variation in the *KCNJ15* gene. The objective of our research was to investigate the potential regulation and control mechanisms of the *KCNJ15* CNV and its utility as a reference for goat selection.

## 2. Materials and Methods

### 2.1. Experimental Samples and Trait Records

We investigated the distribution of the *KCNJ15* CNV among five different goat breeds in China: Guizhou black goat (GZB), Guizhou white goat (GZW), Bohuai goat (BH), Huai goat (HH), and Taihang goat (TH). All experiments conducted in this study met the requirements and were guided by the animal welfare law and related policies of the International Faculty Animal Policy and Welfare Committee of Northwest A&F University. The animal use evaluation committee of Northwest A&F University approved all experiments in this study.

### 2.2. Sample Collection and Genomic DNA Extraction

We collected samples from 706 goats, including ear tissue samples from Taihang goats (n = 98; Jiaozuo city, Henan province) and blood samples from Guizhou black goats (n = 126; Bijie city, Guizhou province), Guizhou white goats (n = 62; Tongren city, Guizhou province), Bohuai goats (n = 294; Yongcheng city, Henan province), and Huai goats (n = 126; Zhoukou city, Henan province). We also recorded data on individual growth traits such as body height, body weight, body length, chest circumference, chest depth, and rump length for subsequent correlation analysis.

In order to identify the distribution of the *KCNJ15* gene’s CNV in the Chinese goat population, five different breeds of Chinese local goats were selected for sampling. Various types of goats inhabit different regions, thrive in different environments, and exhibit different behavioral patterns. All of the goats included in the study were sexually mature female adults ranging in age from 4 to 5 years, with no restrictions on their food source. Additionally, all goats were in good physical condition without disease.

Blood samples were collected from each goat through the jugular vein using a vacuum tube without the need for anesthesia or euthanasia in any of the groups. All goats were in good health and physical condition. The blood samples were placed in centrifuge tubes and stored on ice before being sent to the laboratory. The ear tissue samples were frozen at a medium temperature and kept at −40 °C until use. The phenol–chloroform method was used to extract genomic DNA [14].

### 2.3. Genomic DNA Identification and Primer Test

According to existing research results, we confirmed that there was a CNV fragment of the *KCNJ15* gene in the Chinese goat genome [1,10,11,15,16]. The reference sequence for the candidate region of the target gene’s copy number variant was obtained from the publication of the bovine *KCNJ15* gene (NC_030808.1) in the National Center for Biotechnology Information (NCBI) database. Resequencing analysis was performed on the genomic region Chr1: 150,479,082-150,489,041. The NCBI assembly was ARS1.1 (GCF_001704415.1). Via query and analysis on NCBI, we designed unique primers for the autologous region of the *KCNJ15* gene, as well as designed *MC1R* primers as a reference gene that uses California bio soft international primers (Table 1). At the same time, real-time quantitative polymerase chain reaction (qPCR) was used to draw amplification curves and dissolution peaks to determine whether the primers were appropriate. As shown in Figure 1, the sample curves were consistent. There were no primer dimers or non-specific amplification products.

### 2.4. Detection of the CNV of the Goat KCNJ15 Gene

We used qPCR to determine the potential CNV of the study’s goats. Three genomic DNA qPCR experiments were performed using SYBR Green. The amplification system was performed in a 10 μL reaction system requiring 0.5 μL of upstream and downstream primers, 5 μL of the SYBR premix (Genstar, Beijing, China), and 25 ng of the DNA sample. At the same time, this system carried out a subsequent cycle of 95 °C for 10 min, 95 °C for 15 s, and then 60 °C for 40 s, which was repeated 39 times. At the end of the amplification, the cycle was started at 65 °C and increased by 0.5 °C every 0.05 s until reaching 95 °C to analyze the melting curve.

### 2.5. Statistical Analysis of the Data

The CNV of the *KCNJ15* gene was calculated using 2 × 2^−ΔΔCt^ [17]. The precise calculation technique used was ΔΔCt = ΔCt (experimental group)—ΔCt (reference group), where ΔCt (experimental group) is the result of subtracting Ct (experimental group internal reference gene) from Ct (experimental group target gene), and ΔCt (reference group) is the result of subtracting Ct (reference group internal reference gene) from Ct (reference group target gene). The cycle threshold (Ct) signifies the number of amplification cycles required for the fluorescence signal of the amplified product to reach a predetermined threshold during PCR amplification. The Ct values obtained via qPCR were then utilized to determine the copy number of each individual using the 2 × 2^−ΔΔCt^ method. Subsequently, the values of 2^−ΔΔCt^ and 2 × 2^−ΔΔCt^ were calculated and rounded to assign different individual copy numbers to three types: deletion type (CN = 0 and CN = 1), normal type (CN = 2), and duplication type (CN = 3, CN = 4, and CN ≥ 5).

We employed the general linear model in SPSS (version 18.0; SPSS, Inc., Chicago, IL, USA) to investigate the relationships between the *KCNJ15* CNV and growth traits in the GZH, GZW, BH, HH, and TH breeds. The significance level for this experiment was set at *p* < 0.05. In the data processing, correlation analysis was carried out according to the different factors affecting body shape. Because of the fixed factors such as genetic effect and age, single factor analysis of variance (ANOVA) was used to simplify the analysis. Due to the necessity of minimizing the influence effects and other pertinent considerations, a simplified model was employed as follows for the analysis:*Y_ijk_* = *μ* + *A_i_* + *CNV_j_* + *E_ijk_*
where *Y_ijk_* is the observed growth traits, *μ* is the population average, *A_i_* is the age of each individual, *CNV_j_* is the influence of the jth CNV region of the *KCNJ15* gene, and *E_ijk_* is a random error [18,19,20]. The LSD multiple comparison test was employed to assess the variations among the different groups of data, and the outcomes are reported as mean values ± standard error (SE).

## 3. Results

### 3.1. Distribution of the KCNJ15 Gene CNV in the Five Breeds of Goat

We selected TH, GZW, HH, GZB, and BH to study the distribution of this CNV in different kinds of goats. The Ct value was used for quantitative analysis, and the amplification curve (Figure 1) was then used to determine primer specificity. The three copy numbers (duplication, loss, and normal) were divided into >2 copies, <2 copies, and 2 copies ×2^−Δct^ scans. The results showed the copy number variation in the *KCNJ15* gene in the five goat varieties (Figure 2). The copy number polymorphism frequency of the five goat breeds showed that most of the goats possessed the duplication type (Figure 3).

### 3.2. Effect of the KCNJ15 CNV on the Observed Traits in the Different Breeds of Goats

Numerous recent studies have identified correlations between animal economic traits and copy number variations. We conducted an analysis to examine the association between the CNV types of the *KCNJ15* gene and the economic traits across the five goat breeds using general linear models. According to the results in Table 2, Table 3, Table 4, Table 5 and Table 6, the CNV of the *KCNJ15* gene had a significant effect on the circumference of the cannon bone in the GZB breed (*p* < 0.05), and normal individuals had better growth traits. In the GZW breed, the CNV had a significant effect on body weight and chest circumference (*p* < 0.05). Specifically, individuals with a CNV of 3 displayed superior growth traits. In the HH breed, the CNV had a significant effect on body slanting length (*p* < 0.05). Specifically, individuals with duplication displayed superior growth traits. There was no significant effect of the CNV of the *KCNJ15* gene on the BH breed. In the TH breed, the CNV had a significant effect on body height and body weight (*p* < 0.05). Specifically, normal individuals displayed superior growth traits.

## 4. Discussion

CNVs, which are mainly deletion and duplication at the submicroscopic level, widely exist in mammals and plants and regulate organisms at the genetic level. Studies have shown that the genetic variation in many species is closely related to their economic and growth traits. Numerous prior studies have demonstrated the association between CNVs in multiple genes and growth traits in goats. For instance, an investigation involving 569 goats revealed significant effects of CNVs in the PIGY gene on goat growth traits [21]. Similarly, the CNV of the *SHE* gene was found to be related to economic traits in 750 goats [22]. Furthermore, a study involving 515 goats reported that the CNV of the *MLLT10* gene exerted a substantial influence on growth traits, including hip width [18].

*KCNJ15* is a member of the inward rectifying potassium channel family (KIR), also known as KIR4.2 [9]. The *KCNJ15* gene has three main isomers, but it encodes the same protein. The *KCNJ15* protein is a complete cell membrane protein; its function is more inclined to make potassium ions flow into cells, thus affecting the body’s immune and nervous systems. Therefore, the *KCNJ15* gene has an important application value in goat breeding. We examined the CNV types of the *KCNJ15* gene in a sample of 706 individuals from five different goat breeds. Correlation analysis was conducted to investigate the relationship between these CNV types and the growth traits of the goats. The results revealed variations in the CNV types of the *KCNJ15* gene across the different goat breeds. There was no significant correlation between the CNV of the *KCNJ15* gene and growth traits in BH and HH. However, in TH, GZW, and GZB, the deletion type was dominant, and its growth and development-related traits were also significantly affected. Among the three copy number variants, the duplication type was the most common. As an excellent goat variety in China, the performance of the *KCNJ15* gene CNV in BH and HH goats is completely different from that of the other three. BH goats are a Chinese hybrid, a cross between Henan Huai and Boer goats. This may have been caused by the different genetic relationships and living environments between goat breeds, which need to be further studied in combination with blood relationships.

Given the association between the CNV of the *KCNJ15* gene and goat growth traits, it is posited that the *KCNJ15* gene significantly influences goat growth. Nonetheless, the precise molecular mechanisms underlying this effect, including DNA mutations, translocations, amplifications, and deletions, are currently unknown [23,24]. These diverse genetic alterations have the potential to induce CNVs. To obtain a comprehensive understanding, additional research focusing on these mechanisms is imperative.

Chinese goats have significant characteristics such as tolerance to roughage, strong disease resistance, strong adaptability, stable inheritance, and excellent meat quality. However, the limitations of slow growth and development have hindered the development of China’s goat breeding industry. With the booming development of the Chinese economy, the demand for goats and their by-products in the domestic market has sharply increased. Therefore, it is imperative to improve the production performance and efficiency of Chinese goats. The key points of this research may provide important data for the genetic improvement in Chinese goats, thereby improving their growth ability.

## 5. Conclusions

This study was the first to investigate the CNV of *KCNJ15* in five different goat breeds in China. Our study successfully identified the distribution patterns of the CNV in *KCNJ15* among the five different goat breeds in China and further investigated the correlation between specific CNV types and growth traits in GZB, GZW, TH, HH, and BH goats. The research results indicated that there are significant differences in the distribution of the *KCNJ15* CNV among different goat breeds. In addition, strict statistical analysis also determined that various CNV types have a significant impact on growth traits. Therefore, it is recommended to consider incorporating CNV gain types in breeding plans aimed at growth traits in order to utilize the potential of *KCNJ15*. Our study provides preliminary evidence to support the functional significance of the *KCNJ15* CNV in a wider range of goat breeds. This discovery may provide a new perspective on the potential use of CNVs as promising molecular markers in animal breeding.

In summation, we detected the CNV of the *KCNJ15* gene in different goat breeds in China, which distinctly showed that the degree of variation affected the growth and development of goats to a certain extent, and correlation analysis confirmed this conclusion. Our research demonstrated the effects of the *KCNJ15* CNV in some breeds of Chinese goat for the first time in order to provide evidence for the CNV of the *KCNJ15* gene as a potential factor for goat growth traits. The physiological regulation mechanism of the goat *KCNJ15* gene needs further study. In addition, in order to verify the correlation between the *KCNJ15* gene and goat growth traits, further research and more goat breeds are needed.

## Figures and Tables

**Figure 1 genes-15-00250-f001:**
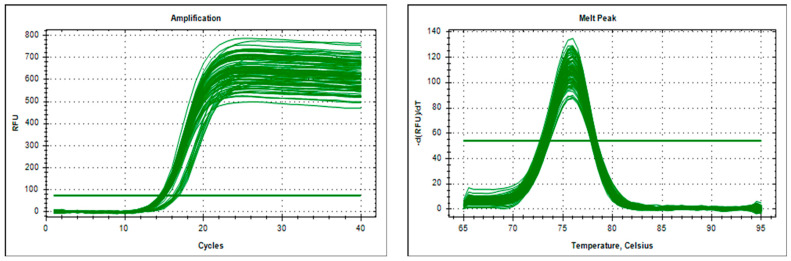
The specific primers of the *KCNJ15* gene.

**Figure 2 genes-15-00250-f002:**
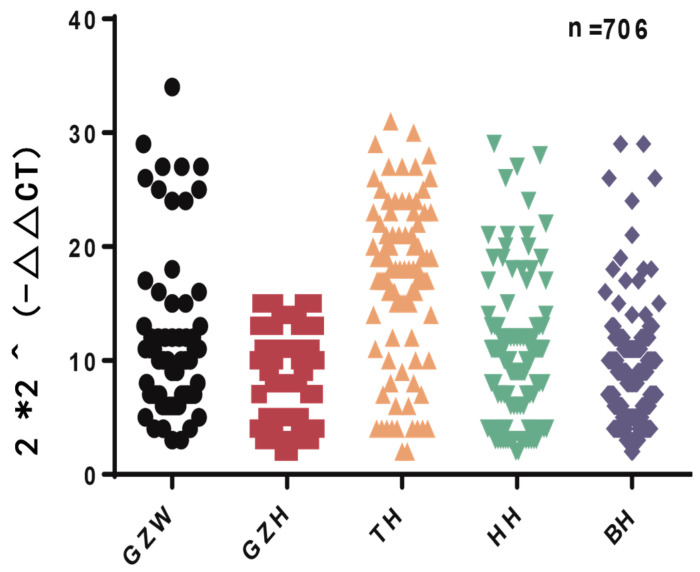
The distribution of the *KCNJ15* CNV among the different species. GZW, Guizhou white goat; GZB, Guizhou black goat; TH, Taihang goat; HH, Huai goat; BH, Bohuai goat.

**Figure 3 genes-15-00250-f003:**
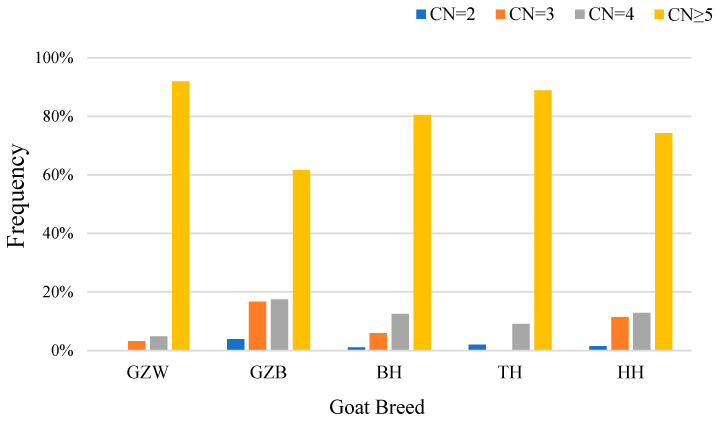
The frequency of the CN of the *KCNJ15* CNV in the five goat breeds. GZW, Guizhou white goat; GZB, Guizhou black goat; TH, Taihang goat; HH, Huai goat; BH, Bohuai goat.

**Table 1 genes-15-00250-t001:** Primer information in the CNV.

Gene	Sequences (5′–3′)	Amplification Length (bp)	Tm (°C)
*KCNJ15*	F1: CCGCTTTCTTGTGAACTTCCAT	80	60
	R1: CTGTCACGTGTCCAGAAAGTAGA		
*MC1R*	F2: GGGCAGTCCCTTGACAAAGA	129	60
	R2: ATCTCCCCAGCCTCCTCATT		

F, forward primer; R, reverse primer.

**Table 2 genes-15-00250-t002:** Association analysis between the copy number variation in the *KCNJ15* gene and growth traits in Guizhou black goats.

Growth Traits	CNV Type (Mean ± SE)				*p*-Value
	Normal	Duplication			
	CN = 2 (n = 5)	CN = 3 (n = 21)	CN = 4 (n = 22)	CN ≥ 5 (n = 78)	
Body weight (kg)	31.50 ± 1.40	27.46 ± 1.74	27.07 ± 1.05	26.85 ± 0.82	0.541
Withers height (cm)	63.40 ± 1.54	59.43 ± 1.31	60.91 ± 0.93	58.39 ± 0.70	0.113
Body length (cm)	61.40 ± 2.08	62.14 ± 1.68	60.95 ± 1.17	60.44 ± 0.74	0.76
Chest measurement (cm)	75.00 ± 0.63	74.81 ± 1.28	73.61 ± 0.95	73.45 ± 0.80	0.807
Circumference of the cannon bone (cm)	9.40 ± 0.25 ^a^	4.84 ± 0.93 ^b^	6.85 ± 0.75 ^ab^	6.87 ± 0.40 ^ab^	0.033 *

Notes: Values with different superscripts (^a, b^) and * within the same row differ significantly at *p* < 0.05.

**Table 3 genes-15-00250-t003:** Association analysis between the copy number variation in the *KCNJ15* gene and growth traits in Guizhou white goats.

Growth Traits	CNV Type (Mean ± SE)			*p*-Value
	Duplication			
	CN = 3 (n = 2)	CN = 4 (n = 3)	CN ≥ 5 (n = 57)	
Body weight (kg)	44.34 ± 2.35 ^a^	22.19 ± 2.79 ^b^	26.96 ± 0.97 ^ab^	0.024 *
Chest measurement (cm)	85.50 ± 1.50 ^a^	70 ± 1.50 ^b^	70.77 ± 0.92 ^b^	0.040 *

Notes: Values with different superscripts (^a, b^) and * within the same row differ significantly at *p* < 0.05.

**Table 4 genes-15-00250-t004:** Association analysis between the copy number variation in the *KCNJ15* gene and growth traits in Bohuai goats.

Growth Traits	CNV Type (Mean ± SE)				*p*-Value
	Normal	Duplication			
	CN = 2 (n = 5)	CN = 3 (n = 27)	CN = 4 (n = 37)	CN ≥ 5 (n = 225)	
Withers height (cm)	64.33 ± 2.67	65.85 ± 2.05	68.78 ± 0.64	67.84 ± 0.39	0.263
Body length (cm)	67.00 ± 3.51	72.29 ± 2.35	76.75 ± 1.17	75.73 ± 0.64	0.159
Circumference of the cannon bone (cm)	9.00 ± 0.58	10.02 ± 0.26	10.26 ± 0.16	10.09 ± 0.64	0.179
Chest measurement (cm)	78.33 ± 3.76	83.88 ± 3.00	92.21 ± 1.39	88.95 ± 1.01	0.138
Body weight (kg)	38.58 ± 5.41	49.40 ± 4.30	60.16 ± 3.03	58.63 ± 1.47	0.135

**Table 5 genes-15-00250-t005:** Association analysis between the copy number variation in the *KCNJ15* gene and growth traits in Huai goats.

Growth Traits	CNV Type (Mean ± SE)				*p*-Value
	Normal	Duplication			
	CN = 2 (n = 2)	CN = 3 (n = 15)	CN = 4 (n = 17)	CN ≥ 5 (n = 92)	
Withers height (cm)	51.50 ± 1.50	60.67 ± 1.58	61.12 ± 1.73	62.27 ± 1.55	0.052
Body length (cm)	52.00 ± 3.00 ^b^	60.80 ± 2.18 ^a^	61.03 ± 1.38 ^a^	63.19 ± 0.63 ^a^	0.044 *
Chest measurement (cm)	59.00 ± 2.00	73.37 ± 2.08	73.97 ± 1.76	75.65 ± 0.99	0.074
Circumference of the cannon bone (cm)	7.50 ± 1.50	8.20 ± 0.22	8.44 ± 0.18	9.31 ± 0.78	0.889
Body weight (kg)	16.85 ± 2.10	31.27 ± 2.92	31.60 ± 2.27	34.47 ± 1.00	0.052

Notes: Values with different superscripts (^a, b^) and * within the same row differ significantly at *p* < 0.05.

**Table 6 genes-15-00250-t006:** Association analysis between the copy number variation in the *KCNJ15* gene and growth traits in Taihang goats.

Growth Traits	CNV Type (Mean ± SE)				*p*-Value
	Normal	Duplication			
	CN = 2 (n = 1)	CN = 3 (n = 0)	CN = 4 (n = 9)	CN ≥ 5 (n = 88)	
Body length (cm)	72.6		67.00 ± 0.85	64.27 ± 0.56	0.109
Withers height (cm)	66.1		61.06 ± 0.67	58.22 ± 0.38	0.010 *
Chest measurement (cm)	82		78.98 ± 2.32	72.55 ± 0.86	0.056
Circumference of the cannon bone (cm)	7.5		7.23 ± 0.18	7.41 ± 0.06	0.67
Body weight (kg)	45.2		39.04 ± 2.46	31.15 ± 0.95	0.045 *

Notes: Values * indicate significant differences (*p* < 0.05).

## Data Availability

The original data are available upon request from the corresponding author.

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
