# Peer review of "The Identification of Goat KCNJ15 Gene Copy Number Variation and Its Association with Growth Traits"

_genes, 2024, doi:10.3390/genes15020250_

Round 1
Reviewer 1 Report
Comments and Suggestions for Authors
See the file.

A deep revision of English language is required.
Author Response
Dear reviewers
We would like to thank the referee once more for sparing the time to write so many detailed and useful comments. We have read through comments carefully and have made corrections.
And we have carefully read your comments and responded to them one by one. Please refer to the attachment for specific details. Based on the instructions provided in your letter, we uploaded the file of the revised manuscript.
We would love to thank you for allowing us to resubmit a revised copy of the manuscript and we highly appreciate your time and consideration.
Sincerely.
Jiahao Zhao, Yong-zhen Huang

Reviewer 2 Report
Comments and Suggestions for Authors
Dear Authors,
I have carefully read your work entitled "Identification of Goat KCNJ15 Gene Copy Number Variation 2 and Its Association with Growth Traits" and believe it may be suitable for publication in Genes.
However, before publication, I think you should consider the suggestions below.
General comments
Looking at figure 5, I have the impression that the data do not distribute according to the normal and therefore in this case the possibility of using ANOVA as a statistical analysis would be lost.
I therefore ask the authors to send me the raw data for CN and the measurements made in the various breeds so that I can verify the correctness of the stasistic analyses, which are the focus of the research and results.
Specific comments
Line 70: You must indicate the reference to OMIN where any phenotypic implications of mutations in this genetic factor can be verified (https://omim.org/entry/602106).
Line 75-77: The reference of where this genotyping can be observed must be included.
Line 78: Reference number 78 is not related to the possible identification of the genotype with horns or without horns.
Line 79: I have carefully read the paper quoted with number 10 and I do not believe that the conclusions are the same as in this paper. The only reported hypothesis of KCNJ15 is that it may have a function in the development of the ovary. Further claims remain to be proven.
Line 79: You have to report the reference of the original PIS paper (Vaiman et al., 1999).
Line 109: The text only states that 706 goats of five different breeds were tested. The authors must include the number of goats belonging to each breed (I did not find this indication throughout the text), and indicate the population parameters such as age and sex. This is important information as the groups must be homogeneousthe fact that they are homogeneous in terms of age and gender must be statistically verified). In this case it is not sufficient to state: All goats are mature and of the same age.
Line 127: Regarding the phrase “According to the results of Comparative Genomic Hybridization(CGH) analysis”: is reference made to a published experiment, preliminary or what?
Line 129: The location of the CNV must be clearly indicated: which chromosome and especially which assembly is being considered?
Line 129: “ranging from 1150347001 to 150541500 BP”. The first number is wrong.
Line 125: reference is not correct.
Line 153: Reference number 10 does not refer to the calculation of areal time results but to DNA extraction.
Line 153: You have to specify the thresholds considered for assigning subjects to 2, 3, 4 and 5 copies (this would make graph 5 more readable)
Line 165-239: I have many doubts about this whole section presenting the bioinformatic analysis of KCNJ15 . I do not think it adds anything to the result of the work and suggest that it be moved to the supplementary results section. Perhaps it would be slightly more interesting to analyse the conservation of this protein in relation to other species (bovidae and non-bovidae).
Line 246: “The copy number polymorphism 246 frequency of five goat breeds showed that most goats showed lost copy number type”. Assuming the correct number of copies is 2, I see no subject with <2 copies in the figure...
Line 256: I have found that table 3 needs a lot of improvement as there are aspects I cannot understand:
a. The number of animals in each category must be indicated.
b. Why do some values in the table show Mean ± SE while others only Mean?
c. For GZB Body weight, GZW Chest circumference, HH Body height and TH Body weight I believe that the value 0 indicates that there are no animals in these categories, so remove 0 and put n.a. (not available)
d. All of the data in the table are significant (p<0.05), however, in a large number of cases there is a lack of significance letters showing who is different from who
e. The caption of the table reports “HB= Henan Bohuai goat”, however, data on this breed are not shown. I guess they are not significant but must be reproduced anyway.
f. The caption of the table reports “n=the number of goat with different CNV types”, but this data is not fully and clearly reported.
Line 286-287: “706 goat breeds”, I think it is wrong: you analysed 706 goats of 5 different breeds.
Line 289: “the deletion type was dominant”, as reported for line 246, I cannot understand this statement.
Author Response

(The authors gave the same response as above.)

Reviewer 3 Report
Comments and Suggestions for Authors
Dear Authors,
The revised manuscript version 'Identification of Goat KCNJ15 Gene Copy Number Variation 2 and Its Association with Growth Traits' is now more complete.
Less technical manuscript additions also need to be done.
Line 78: instead of "et al" write "et al."
Lines 280 and 281: determine the correct writing “KIR“ or “Kir“; “kir4.2.“ or “ Kir4.2.“
Line 329: The abbreviation CN is not clarified in the list of abbreviations.
Line 365 and beyond: It is necessary to edit them according to the instructions to the authors in the list of references. Some references are not written correctly.
Author Response

(The authors gave the same response as above.)

Reviewer 4 Report
Comments and Suggestions for Authors
The paper deals with the analysis of KCNJ15 gene in Chinese goat breeds.
I am a bit discomfited when reading the text.
Methods are not properly described.
At which age were the goats measured.
The characteristics of protein are described in detail, but not eventual differences among CNVs.
Delete rr. 84-93.
Table 3, you write in the legend n=number of goats, but n is not in the table. There is twice body weight. There is not mentioned HB breed.
Give the frequencies in breeds and in the all group analyzed. Give the body measures within breed for different CNVs. Was the effect of the breed significant?
Fig. 6, GZH, BH and HUAI are in the figure, but are not in the legend; conversely GZB, HH and HB are in the legend, but not in the figure.
What is canno round.
You write in the text that the deletion variant was most common, in the Fig. 6 are most frequent CN≥5.
Rr. 285-6, 706 goats were analyzed, not goat breeds.
Formal errors:
Abstract, explain the abbreviations.
Check the formulation in rr. 50-52, 101-103, 324 and in other places.
R. 124, -40.
R. 156, divide into 2 paragraphs, equation, and legend.
R. 240 breeds of goat, not species.
Generally, I suggest remaking of the paper.
Comments on the Quality of English LanguageSome formulations must be corrected.
Author Response

(The authors gave the same response as above.)

Round 2
Reviewer 1 Report
Comments and Suggestions for Authors
No suggestions
Comments on the Quality of English LanguageMinor editing of English language required
Author Response
I sincerely thank you for your approval of the second revision of my manuscript entitled "Identification of copy number variations in the goat KCNJ15 gene and their association with growth traits".
I am grateful for the time and expertise you have dedicated to the review process. At the same time, we have also invited an English editor from MDPI to provide language polishing for the entire text,Your valuable insights have played a pivotal role in enhancing the overall quality of my work. Thank you once again for your invaluable contributions.
Best regards,
Jiahao Zhao, Yong-zhen Huang
Reviewer 2 Report
Comments and Suggestions for Authors
Dear Authors,
I have read your second version and am satisfied with the changes you have made.
I have checked the data and confirm that the statistical analyses were followed correctly.
I only ask you to make the small changes that I indicate below.
Line 119: I do not understand why you refer to the STAT5A gene that is located in a completely different position.
Line 121: The name of the genome assembly related to GCF_001704415.1 is ARS1.1
Line 146-151: I am sorry, but you did not answer my question: when performing a Real Time PCR, you do not obtain whole values (2, 3 or 4) but intermediate values, e.g. 2.5765. My question is: what are the limits of these measurements for classifying animals with 1, 2 3 or more than 4 copies?
Line 191: Delete the following sentence from the caption of all tables (2 to 5): “Values in the same row marked A, B and ** indicate a very significant difference (P < 0.01).”. No significant results for p<0.01 are present.
Line 197-198: In table 3 there are still 0 values to indicate no animals: delete these values.
Line 197-198: Table 3 lacks significance letters in the categories. Also missing for CN=4 measurements is the standard error.
Line 209: In table 5 the letters of significance must be month as superscript.
Author Response
I am grateful for the time and expertise you have dedicated to the review process. and thank you again for acknowledging our modifications, We have made modifications based on your feedback and attached them in the attachment, hoping to answer your question. At the same time, we have also invited an English editor from MDPI to provide language polishing for the entire text,Your valuable insights have played a pivotal role in enhancing the overall quality of my work. Thank you once again for your invaluable contributions.
Best regards,
Jiahao Zhao, Yong-zhen Huang

Reviewer 4 Report
Comments and Suggestions for Authors
The paper was thoroughly remade. Just some imperfections remain.
Add abbreviations of breeds into Abstract.
Fig. 3 is twice(?).
Author Response

(The authors gave the same response as above.)
